# Interpretation of Geological Features and Volcanic Activity in the Tsiolkovsky Region of the Moon

Ying Wang [1,2], Xiaozhong Ding [1,2,*], Jian Chen [3], Kunying Han [1,2], Chenglong Shi [1,2], Ming Jin [1,2], Liwei Liu [1,2], Xinbao Liu [1,2] and Jiayin Deng [4,5]

1. Institute of Geology, Chinese Academy of Geology Sciences, Beijing 100037, China; wymaggie0312@163.com (Y.W.); kunyinghan@163.com (K.H.); scl0322@foxmail.com (C.S.); jinmingjsh@163.com (M.J.); 18810598150@163.com (L.L.); 2004220020@email.cugb.edu.cn (X.L.)
2. National Research Center of Geological Mapping, China Geological Survey, Beijing 100037, China
3. Institute of Space Sciences, Shandong University, Weihai 264209, China; merchenj@sdu.edu.cn
4. School of Civil Engineering and Architecture, Henan University of Science and Technology, Luoyang 471023, China; dengjy.20b@igsnrr.ac.cn
5. State Key Laboratory of Resources and Environmental Information System, Institute of Geographic and Natural Resources Research, Chinese Academy of Sciences, Beijing 100101, China
* Correspondence: xiaozhongding@sina.com; Tel.: +86-136-6113-9832

**Abstract:** The Tsiolkovsky crater is located on the farside of the Moon. It formed in the late Imbrian epoch and was filled with a large area of mare basalts. Multisource remote sensing data are used to interpret the geological features of the Tsiolkovsky area. Compared with previous studies, new remote sensing data and a chronological model based on crater size–frequency distribution are used to further refine the stratigraphic units and determine the absolute ages of the mare basalt units. The evolution of volcanic activity in this crater is discussed. The results are as follows: Abundances of major elements, Th, and silicate minerals suggest that the mare basalt in the crater floor is not a uniform unit but rather nine units with different compositions. The nine basalt units are divided into two episodes of volcanic activity: The first occurred at 3.5–3.7 Ga, when highly evolved lava erupted at the crater floor at a large scale; the second occurred at ~3.4 Ga, when a small area of more primitive lava extended to the northern portion of the crater floor.

**Keywords:** Tsiolkovsky crater; mare basalt; volcanic activity; geologic characteristics; age

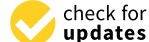



## 1. Introduction

The Moon is the closest celestial body to the Earth and is the window for humans to explore the universe. The gravity on the surface of the Moon is only one-sixth that of the Earth's surface, and using the Moon as a base for space science research stations and deep-space exploration activities will help save fuel and thus reduce the cost of space exploration missions. In addition, the geological structure of the lunar surface is relatively stable, there is no interference such as the atmosphere, and other celestial bodies in the universe can be more clearly and accurately observed on the Moon. However, there are also some problems, such as the poorly known stability of the regolith, abrasive aspects of the regolith grains, and radiation exposure. The great importance of lunar exploration has been recognized by various countries. Luna missions from the Soviet Union and Apollo missions from the United States were conducted to obtain samples from the Moon. In recent years, China has vigorously developed a lunar exploration project, and the Chang'e-5 mission collected 1731 g of sample from the northwestern region of the lunar Oceanus Procellarum [1]. Due to the limitations of aerospace technology, it is very difficult to sample extraterrestrial bodies. Furthermore, a planetary exploration mission needs a lot of manpower and material preparation work, which is time-consuming and expensive. There may also be difficult technical problems in collecting and returning samples. The lack of

samples and the poor global representativeness of the currently available samples limit the understanding of the lunar materials. In contrast, remote sensing technology can more easily and quickly obtain information of the planet's surface from orbital platforms. Remote sensing methodologies are combined with returned samples to study more extensive lunar surfaces. The exogenic impact process greatly influences the morphology of the lunar surface. Many studies have been performed on the nearside of the Moon, but fewer have been performed on the farside [2–4]. The volcanic processes on the farside of the Moon are also significantly different from those on the nearside [3]. The Tsiolkovsky crater on the farside of the Moon is a typical lunar complex crater with a central peak and a large area of mare basalts on its floor. A comprehensive analysis of volcanic activity in the Tsiolkovsky crater is conducted in this study.

Due to technical limitations, there have been a lack of lunar samples directly returned from the Tsiolkovsky area, and most previous remote sensing studies in this area have focused on magma sources and the basalt thickness, age, mineralogy, and other aspects. In 1973, Gornitz et al. [5] analyzed the mare-filled impact structure of the Tsiolkovsky crater using Lunar Orbiter and Apollo images and concluded that the lava in the crater originated from internal melting. Later, Walker et al. [6] used lunar topographic orthophoto maps to calculate a thickness of 1.75 km and a volume of $3.6 \times 10^4$ km$^3$ for basalts in the Tsiolkovsky crater. As lunar exploration datasets are continuously updated, Tyrie et al. [7] used Apollo 15 panoramic camera photographs to date the basalt via the crater size–frequency distribution method, and the result was $3.51 \pm 0.1$ Ga. Later, Salih et al. [8] used updated Kaguya Terrain Camera (TC) data to determine that the age of the Tsiolkovsky crater floor is typically 3.2–3.3 Ga. In terms of the mineral distribution, Heather et al. [9] used Clementine mosaics to determine that the central peak is feldspathic with very small, locally olivine-rich outcrops. In addition, Othake et al. [10] found that there is an outcrop of pure anorthosite with a ~100% plagioclase content at the top of the central peak. Matsunaga et al. [11] suggested that the central peak is not olivine-rich but rather a mixture of plagioclase and pyroxene. Previous studies tended to focus on one aspect of the region or used older data. Using the latest remote sensing data, a comprehensive geological study of the relatively young central peak crater with a large amount of mare basalt, which is a special geological feature on the farside of the Moon, needs to be performed.

In this paper, multisource remote sensing data, such as lunar orbiter laser altimeter (LOLA) data, lunar reconnaissance orbiter camera wide-angle camera (LROC WAC), Chandrayaan–1 Moon Mineralogy Mapper (M$^3$) reflectance data, lunar crustal thickness data models [12], and lunar prospector Th abundance data are used. The regional geological background, topographic features, mineral distribution, structural features, and ages of the mare basalts in the Tsiolkovsky crater are analyzed in detail, and the evolutionary history of volcanic activity in the crater is discussed.

## 2. Geologic Context of the Tsiolkovsky Crater

The Tsiolkovsky crater formed in the late Imbrian epoch at approximately 3.8 Ga [6]. It is one of the youngest complex craters on the farside of the Moon. The Tsiolkovsky crater has a complete structure (Figure 1) with an average rim diameter of approximately 180 km [13] and an average depth of approximately 4 km. It is centered at 129°E, 20°S and has a very prominent central peak approximately 1.7 km above the crater floor. The ejecta material of this crater is radially distributed around the rim of the crater, which has an irregular edge and is a typical central peak crater on the farside of the Moon. The crater has a smooth and dark floor covered with mare basalts. The crater is located in the low-latitude region of the southern hemisphere and is part of the anorthositic highlands tectonic unit [14]. Before the Tsiolkovsky impact, the Aitkenian Fermi basin had already produced thick ejecta materials at the target site, after which the Patsaev, Waterman, and Neujmin impact craters were formed in the Nectarian period and Litke impact crater was formed in the early Imbrian epoch; these craters also contributed to the target materials of the Tsiolkovsky impact.

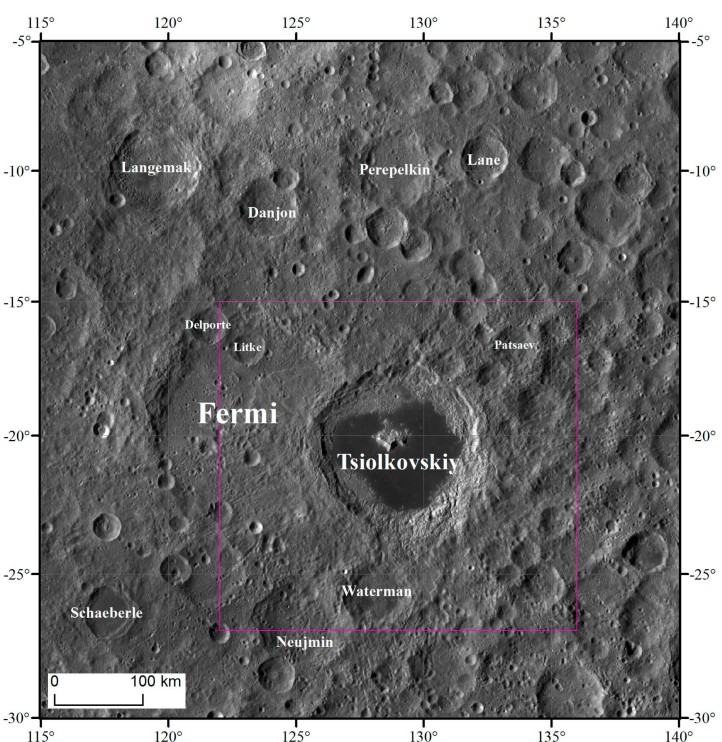

**Figure 1.** Location of Tsiolkovsky crater.

## 3. Data and Methods

### 3.1. Data

For the morphologic and topographic analyses, we used WAC mosaic and LOLA digital elevation model (DEM) data, both of which were acquired by instruments mounted on the Lunar Reconnaissance Orbiter (LRO). The WAC is a multispectral, wide–angle camera that obtains ultraviolet and visible spectral data with a spatial resolution of 100 m/pixel for the lunar global surface. The LOLA is an altimetry system that derives a topographic product with a resolution of 118 m/pixel for the lunar global surface. The Tsiolkovsky region is ~200 km wide; hence, the WAC mosaic and DEM with a spatial resolution of ~100 m (~0.1 mm in the map with a scale of 1:1,000,000) are sufficient for the analysis of topographic features.

For the chemical compositional analysis, we employed four types of elemental abundance maps: $TiO_2$, FeO, Mg#, and Th maps. A $TiO_2$ abundance map was produced by Sato et al. [15] using the correlation between the global WAC 321 nm/415 nm band ratio and the $TiO_2$ content of soil samples collected from geologically homogeneous areas of the Apollo and Luna landing sites. The FeO abundance map was obtained by Lemelin et al. [16] based on the Kaguya multiband imager data. A Th abundance map was acquired by a Lunar Prospector Gamma-ray Spectrometer with a pixel resolution of $0.5° \times 0.5°$ [17,18]. The Mg# map was derived by Liang Zhang et al. [19] using a one-dimensional convolutional neural network (1D–CNN) algorithm combined with the Kaguya multiband imager and geochemical information from Chang'e-5 samples. The lunar crustal thickness (model 4) calculated by Wieczorek et al. [12] based on the Gravity Recovery and Interior Laboratory (GRAIL) gravity field data was also analyzed.

For the spectral characteristics and the mineralogical analysis, we used the $M^3$ hyperspectral imaging data. The Moon-Mineralogy Mapper ($M^3$) is an advanced spectrometer developed by the National Aeronautics and Space Administration (NASA) and is a guest sensor aboard the Indian Chandrayaan–1. Its spectral range is in the visible and near-infrared bands (0.46–3.0 μm), which cover the absorption bands of lunar mafic minerals [20]. The $M^3$ data are divided into two categories: low–resolution global mode data with a resolution of 140–280 m/pixel and nearly global coverage, including 85 bands; and target mode

data with a resolution of 70–140 m/pixel and containing 260 bands, covering a small range of the lunar surface [21]. The M³ data cover a total of 2 optical periods, and the data for the mineral analysis in the study area are products of the first (OP1B) and the second (OP2C) optical periods (Table 1).

**Table 1.** Information of M³ datasets used in this study.

| Data File Name | Date | Orbit Altitude (km) | Optical Period | Resolution (m/pixel) | Phase Angle Range (°) |
|---|---|---|---|---|---|
| m3g20090127t031145_V01_RFL | 20090127 | 100 | OP1b | 140 | 35–90 |
| m3g20090529t061013_V01_RFL | 20090529 | 200 | OP2c | 280 | 0–100 |
| m3g20090529t100749_V01_RFL | 20090529 | 200 | OP2c | 280 | 0–100 |
| m3g20090529t143509_V01_RFL | 20090529 | 200 | OP2c | 280 | 0–100 |
| m3g20090529t183825_V01_RFL | 20090529 | 200 | OP2c | 280 | 0–100 |
| m3g20090529t230608_V01_RFL | 20090529 | 200 | OP2c | 280 | 0–100 |
| m3g20090530t030925_V01_RFL_ | 20090530 | 200 | OP2c | 280 | 0–100 |
| m3g20090530t073724_V01_RFL_ | 20090530 | 200 | OP2c | 280 | 0–100 |
| m3g20090626t142653_V01_RFL | 20090626 | 200 | OP2c | 280 | 0–100 |
| m3g20090626t182943_V01_RFL_ | 20090626 | 200 | OP2c | 280 | 0–100 |

*3.2. Methods*

The range of the study area is 122–136°E, 15–27°S. We used the WAC mosaic, LOLA DEM, and crustal thickness model [12] to characterize the geological background and topographic features in detail. GIS software (version 10.8), DEM maps, slope maps, and lunar crust thickness maps were used to analyze the topographic features in the study area.

Then, the WAC data, MI data, LP Th abundance data, and Mg# map were integrated with ArcGIS as the operating platform, and the important geological information implied by the elemental distribution in the study area was analyzed in detail. The chemical compositions of basalt units in the study area were analyzed by zonal statistics tools. Zonal statistics tools is in ArcGIS that is mainly used to calculate statistics within a specific area.

On this basis, M³ hyperspectral imaging data were used to identify and map the main minerals in the Tsiolkovsky impact crater. The specific steps were as follows. The M³ Level 2 reflectance data in the study area were corrected by ground truth spectra and geometrically corrected. Then, the M³ images were mosaiced, and spectral data were extracted. Spectral continuum removal was carried out on the extracted spectral curves of the regions of interest, the integrated depth of absorption features (IBD) was calculated, and false color composites of the IBD images and mineral indices were produced. The major mineral types and distributions were determined based on mosaiced M³ images. To comprehensively analyze the basic mineral characteristics of the study area, we used spectral parameters to produce false color (RGB) images. The IBD at the 1000 nm and 2000 nm bands and the reflectance at the 1550 nm band can reflect the distribution of major minerals in the study area, so we used these three parameters as red, green, and blue channels, respectively, to generate an RGB false-color map. The spectral parameters and integrated absorption depth (IBD) were calculated via Formula (1) and Formula (2) [22]. In the formulas, *R* represents the reflectance in the band, *Rc* is the corresponding envelope continuum value in the band, 20 and 40 are the wavelength sampling intervals, and *n* is the number of bands contained in the absorption bands. Few mineral types occur on the Moon, mainly plagioclase, pyroxene, and olivine, and these mineral types have corresponding spectral absorption characteristics. Based on the information from the IBD false–color images, the basalt emplaced at the floor of the Tsiolkovsky impact crater is divided into several units.

$$\text{IBD (1000 nm)} = \sum_{n=0}^{26} \left(1 - \frac{R\,(789 + 20 \times n)}{Rc\,(789 + 20 \times n)}\right) \tag{1}$$

$$\text{IBD (2000 nm)} = \sum_{n=0}^{21} \left(1 - \frac{R\,(1658 + 40 \times n)}{Rc\,(1658 + 40 \times n)}\right) \tag{2}$$

In this study, the chronological function of Yue et al. [23], CraterTools, and Craterstats 2.0 software were used to measure the crater size–frequency distribution of each basalt unit on the floor of the crater [24,25]. Finally, the evolution of volcanic activity in this area was analyzed and discussed based on the topography, chemical composition, spectral characteristics, and age of each unit.

## 4. Results

### 4.1. Topographic Features

The Tsiolkovsky crater is centered at 129°E, 20°S and is located in the anorthositic highlands. The structure of the crater is complete and includes central peak materials, floor materials, wall materials, and continuous ejecta materials. The degree of degradation is relatively low, and the topological features are well preserved. The central peak is approximately 1.7 km higher than the crater floor and is shaped like a 'W'. Most of the surface is smooth and has not been changed by the later impact. In the 1:2.5 M–scale geological map of the Moon by Ji Jinzhu et al. [26], the formation age of the crater is considered late Imbrian. The geological context surrounding the crater is relatively simple, and due to its relatively young age, the impact overlaps several older impact craters. In particular, the Fermi basin is located west of the Tsiolkovsky crater with a diameter of ~230 km [27], and the formations of its eastern basin wall and rim have been modified by the Tsiolkov-sky crater. The Fermi basin formed in the Aitkenian period. Over time, the geological characteristics of the surface have been degraded by space weathering and impacts, but the basin floor formation, basin wall formation, and basin rim formation can still be distinguished. To the south of the Tsiolkovsky crater are the Waterman and Neujmin craters, both of which formed during the Nectarian period [26]. According to the stratigraphic relationship, the Waterman impact crater destroyed the ejecta blanket of the Neujmin impact crater, so the former was younger. To the northeast of Tsiolkovsky are the Lander and Patsaev craters, which also formed during the Nectarian period [26]. The interior of the Patsaev crater is smooth in the image, and the ejecta blanket has been modified by subsequent impact events. It may, therefore, be more exposed to spatial weathering. The Lander crater is better preserved than the Patsaev crater and may, therefore, be younger.

As shown in Figure 2b, the elevations of the research area range from −3238.5 to 4951.5 m, and the relative elevation difference is 8190 m. Overall, the terrain is high in the north and low in the south, and the western part of the crater has the lowest elevation. The image shows that the lowest portion is the landslide, a foliated structure that extends westward into the Fermi basin floor and is very extensive, which is very unique on the Moon. Boyce et al. [27] suggested that this is the zone lacking ejecta due to oblique impact. As shown in Figure 2c, the slopes of the study area range from 0 to 21.7°. The floor of the Tsiolkovsky impact crater is very gentle. The places with large slopes are located at the central peak and the highest position of the crater wall, indicating that fresh rock outcrops may exist.

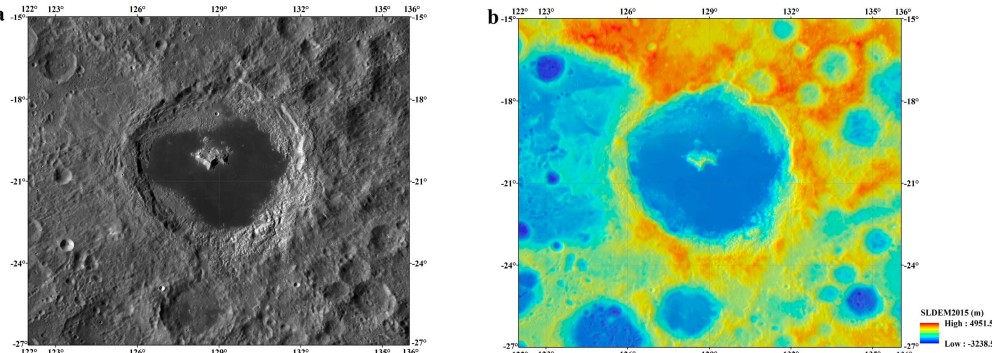

**Figure 2.** *Cont*.

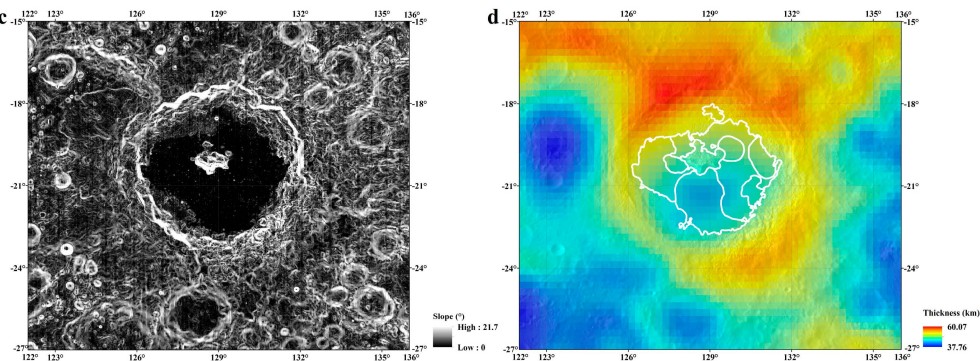

**Figure 2.** Topographic features of the study area. (**a**) WAC image; (**b**) DEM map; (**c**) Slope map; (**d**) Crustal thickness model. WAC image (~100 m/pixel) is superimposed on panels (**b**–**d**) with 75% transparency.

### 4.2. Spectral Characteristics and Mineral Distribution

From the false-color image, it is obvious that the crater floor is not composed of uniform materials. Therefore, according to the different compositions, we divided the whole floor of the Tsiolkovsky crater into a total of 12 units, A–L, as shown in Figure 3. Among them, units A, B, C, E, F, G, H, I, and J are the mare basalt units of the crater floor; units K and L are floor materials that formed by the collapse of the crater wall; and unit D is the central peak material.

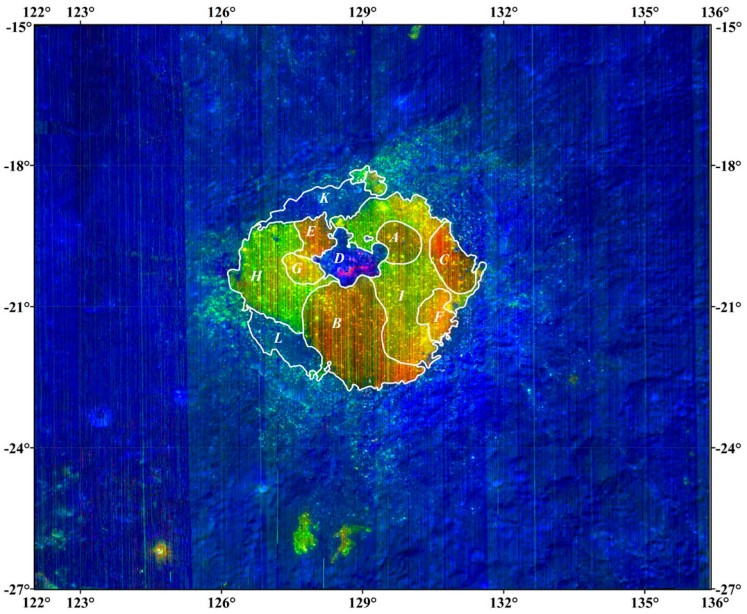

**Figure 3.** Impact crater unit division map.

In these units, we observed the spectral characteristics of pyroxene minerals, crystalline plagioclase, and other minerals. To reduce the effects caused by space weathering, we collected the spectra of fresh outcrops in the study area. To observe the absorption peak centers more clearly, we performed continuum removal with the straight–line segments defined at three points above 750 nm, 1550 nm, and 2500 nm [22]. The original reflectance spectra of the M$^3$ data and the representative spectra after continuum removal are shown in Figure 4.

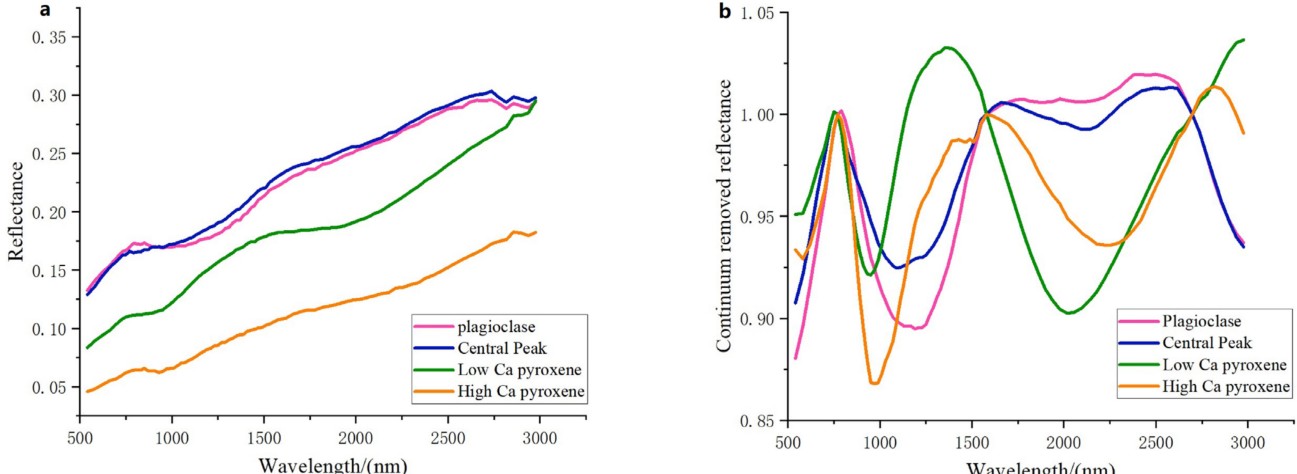

**Figure 4.** (**a**) Representative M$^3$ spectra of the study region; (**b**) Continuum-removed spectra.

Lunar pyroxene minerals are generally divided into three categories: orthopyroxene, low–calcium pyroxene (LCP), and high-calcium pyroxene (HCP). The spectral differences at 1000 nm and 2000 nm are mainly caused by the electronic transition of $Fe^{2+}$. In addition, $Mg^{2+}$ and $Ca^{2+}$ cause changes in the spectral characteristics of pyroxene minerals because they occupy the M1 and M2 positions in the lattice structure, affecting the crystal field environment of $Fe^{2+}$ [28]. In the areas shown in yellow and green tones in the false–color composite, that is, basalt units H, I, G, and J, the absorption characteristics occur at approximately 950–990 nm and 2000–2200 nm, suggesting that these places are rich in low-calcium pyroxene. In addition, similar LCP–rich characteristics exist in the continuous ejecta in the northeast and southwest parts of the crater. The regions shown in the orange hue in Figure 3, namely, basalt units A, B, C, and E, all have absorption characteristics at approximately 970–2300 nm, suggesting that these areas are more enriched with high-calcium pyroxene. No units rich in orthopyroxene minerals were observed in the study area.

We extracted the spectral curve at the central peak of the Tsiolkovsky impact crater, which showed red and purple tones in the false-color image. The spectral features present a relatively broad absorption peak at approximately 1000 nm, the absorption center is approximately 1100–1200 nm, and there is no absorption at approximately 2000 nm. This finding suggests that lithologies rich in olivine or crystalline plagioclase may be present in this area. They are not widely distributed in the Tsiolkovsky crater, mainly appearing to the southeast of the central peak and at a higher elevation. However, the presence of olivine in these areas is questionable due to the slightly blurred characteristics for identifying olivine. Tompkins et al. [29] used Clementine ultraviolet–visible camera multispectral data to propose the presence of olivine on the central peak of the Tsiolkovsky crater, while Yamamoto et al. [30] argued that the spectral bands of the Clementine dataset are discontinuous and that the spectral coverage is limited; moreover, there may be misidentification of minerals. Using SP data with a larger spectral range and smaller sampling intervals, Yamamoto et al. [30] claimed that the olivine spectral signature that emerged from the central peak of the Tsiolkovsky crater was actually a mixture of plagioclase and pyroxene.

Plagioclase, which is widely distributed on the Moon, lacks absorption characteristics at approximately 1250 nm due to long–term space weathering and has high reflectivity, so it appears blue on the false–color map. The mineral is widely distributed across much of the central peak, the northwestern crater wall, and the southeastern crater wall. The main features of crystalline plagioclase are higher reflectance and a characteristic absorption at 1250 nm. We observed outcrops of crystalline plagioclase on the central peak and the southeastern crater wall. These results are consistent with those of Ohtake et al. [10] and Yamamoto et al. [31].

### 4.3. Chemical Composition and Lunar Crustal Thickness Analysis

The abundances of the main elements on the lunar surface have always been an important research topic in lunar science, and many studies have been conducted by predecessors, especially quantitative mapping of the abundances of important elements, such as iron and titanium. Variations in the abundance of elements play an important role in classifying lunar surface rock types and interpreting the regional geological evolution. As shown in Figure 5b, the FeO content in most areas of the floor of the Tsiolkovsky impact crater is homogenous, ranging from 13.53 wt.% to 21.93 wt.%, while the FeO content in most areas of unit J ranges from 10 wt.% to 13.53 wt.%. According to Chen et al.'s classification scheme [32] of lunar surface lithologies, these areas are classified as mare basalt units. An analysis of the $TiO_2$ content can further classify the types of mare basalts in these units. As shown in Figure 5a, compared with the distribution of the FeO content, the $TiO_2$ content of the crater floor mare basalt is heterogeneous and presents a patchy pattern. The $TiO_2$ contents in the northern and southeastern parts of unit I are very low, and the $TiO_2$ content in the remaining area of unit F is extremely low, except in sporadic locations where the $TiO_2$ contents are between 1.5 wt.% and 4.5 wt.%. Unit H has extremely low $TiO_2$ contents in small areas to the south and northeast. The $TiO_2$ content is lower than the detection limit in the eastern part of unit G. The mare basalts on the floor of the Tsiolkovsky crater generally have $TiO_2$ contents between 1.5 wt.% and 4.5 wt.% and are low-titanium basalts.

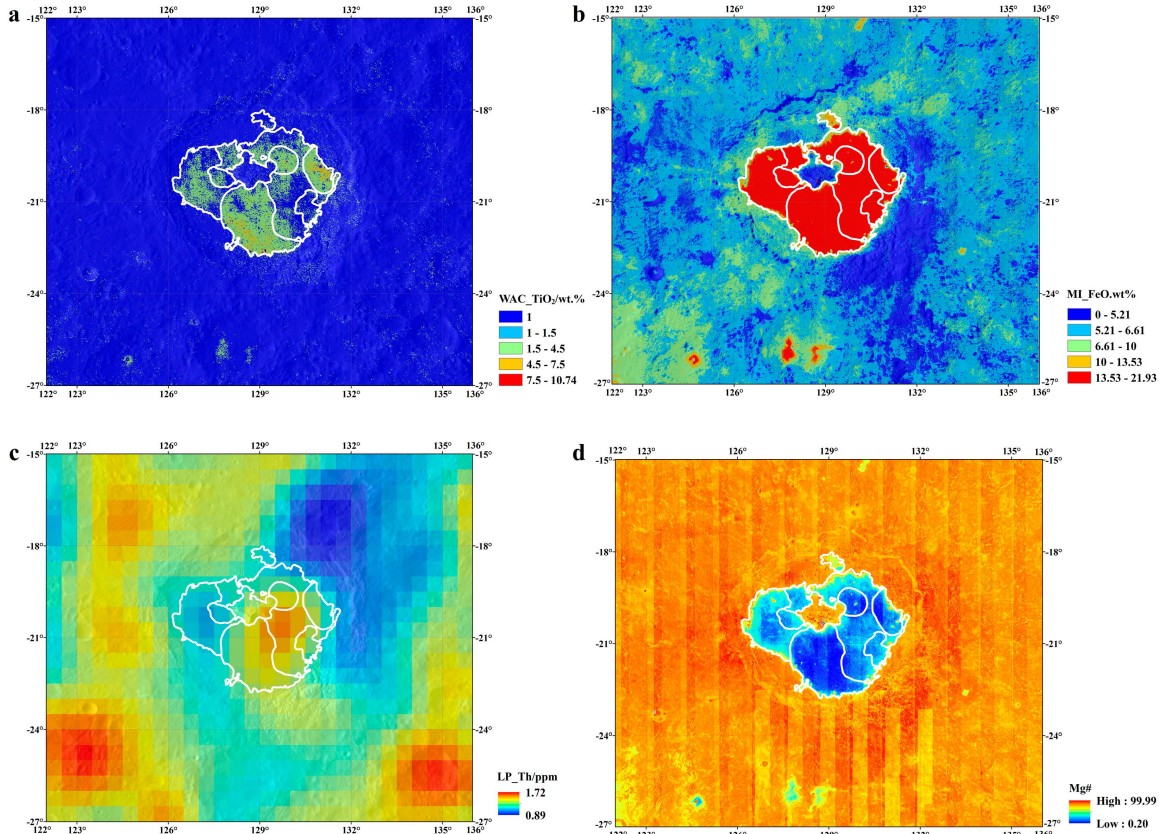

**Figure 5.** Chemical maps of the study area. (**a**) WAC $TiO_2$ abundance (wt.%) map; (**b**) MI FeO abundance (wt.%) map; (**c**) Th map from the LP GRS data; (**d**) Mg# ($100 \times MgO/(MgO + FeO)$) map using data from Liang Zhang et al. [19].

Th is a common heat-producing element on the Moon, and its content can usually represent the amount of other incompatible and heat-producing elements. As shown in Figure 5c, the spatial distribution of the Th content in the study area is inhomogeneous. The southern part of unit I and the eastern part of unit B have high Th contents with local

maxima of 1.52 ppm to 1.72 ppm, and most areas have Th contents between 1.27 ppm and 1.52 ppm. The Th contents in units A, B, C, I, and F are relatively high in the basalt unit present on the whole crater floor, which may have heated the underlying lunar mantle and provided melting conditions for basaltic magmas to erupt on the lunar surface. In addition to heat–producing elements, the thickness of the lunar crust is usually one of the important conditions for basalt to erupt on the lunar surface. As shown in Figure 2d, the thicknesses of the lunar crust of unit B are the lowest at the crater floor, ranging from 37.76 km to 45.17 km. These thicknesses are followed by the southern area of unit I, which is approximately 48.5 km thick. The thicknesses of the lunar crust of the remaining units range from 48.5 km to 56 km. The thinner the lunar crust is, the easier the eruption of magma, which may also be one of the important reasons for the large extent of lava flooding in unit B and unit I. The Mg# can represent the degree of magma evolution. Figure 5d shows that the Mg# value of the basalt at the crater floor is also heterogeneous. Unit B has a low Mg# value, followed by unit A and unit C, and the remaining units have high Mg# values.

*4.4. Chronology*

Many studies have been performed on the age of this crater and mare basalts in this crater. First, Boyce and Johnson [33] suggested that the age of the Tsiolkovsky basalt is 3.8 Ga. As detection data have been continuously updated, Walker et al. [6] mapped 4717 impact craters in an area of 7384 km$^2$ and calculated the basalt age to be 3.8 Ga, i.e., Imbrian. In 1988, Tyrie et al. [7] randomly selected 85 basalt regions, excluding those covered by volcanogenic and secondary impacts, and mapped 12604 impact craters with diameters ranging from 70 m to 1 km, deriving a dating result of 3.51 ± 0.1 Ga. Salih et al. [8] used Kaguya Terrain Camera images with a resolution of 7.4 m per pixel to select an area of 100 km$^2$ on the basalt unit at the crater floor as a reference. The selected diameters of the craters ranged from 128 to 1000 m for dating, and the ages were 3.2–3.3 Ga with the youngest being 2.9 Ga and the oldest being 3.6 Ga in local areas. Boyce et al. [34] estimated the age of the crater and its landslide to be 3.6 Ga and the age of the surrounding Aitkenian highlands to be 4.05 Ga. In the same year, Greenhagen et al. [35] studied the impact crater size–frequency distribution at the northwestern rim of the crater, measuring a model age of 3.8 ± 0.1 Ga. They identified 119 impact craters with diameters greater than 500 m in an area of 9191 km$^2$ with a calculated age of $3.32^{+0.06}_{-0.08}$ Ga. Among them, 43 impact craters with diameters greater than 200 m were mapped in an area of 1115 km$^2$ west of the crater floor. A similar age was derived, i.e., $3.25^{+0.11}_{-0.23}$ Ga [35].

By summarizing the results of previous studies, we found that they usually dated the entire basalt region on the crater floor. However, after compositional analyses in the study area, it is found that the basalt units at the floor of the Tsiolkovsky crater are not uniform but consist of multiple units with different compositions, suggesting that there may be multiple stages of volcanic activity. To more clearly define the sequence of volcanic activity, crater size–frequency distribution measurements were performed for each basalt unit.

We use WAC images as the base map to demarcate the basalt units on the floor of the Tsiolkovsky crater and use M$^3$ false-color composites to identify and map the distribution of the basalt units. The basalt at the floor of the crater is subdivided into nine units, i.e., A, B, C, E, F, G, H, I, and J, according to their distinct compositions. The shape of the basalt units is extremely irregular; hence, the rectangular area with the largest area within the irregular unit is drawn as the dating area to eliminate the influence of area uncertainty. Unit J is an isolated unit, so the irregular polygon is directly used as the dating area of the unit. In this paper, the cumulative frequency of impact craters with diameters greater than 0.4 km was counted, the impact crater cumulative frequency curve was fitted by the least squares method [25], and the absolute model age was obtained by incorporating the lunar chronology function [23]. In this study, the surface ages of each unit were calculated and compared with previous stratigraphic unit division and dating results, revealing that the basalts that developed in the Tsiolkovsky crater were all late Imbrian rocks. The ages of the

nine units, namely, A, B, C, E, F, G, H, I, and J, are 3.57 Ga, 3.63 Ga, 3.66 Ga, 3.52 Ga, 3.64 Ga, 3.62 Ga, 3.64 Ga, 3.65 Ga, and 3.41 Ga, respectively. The results are shown in Table 2. The CSFD plots of each unit produced by Craterstats software (version 2.0) are shown in Figure 6.

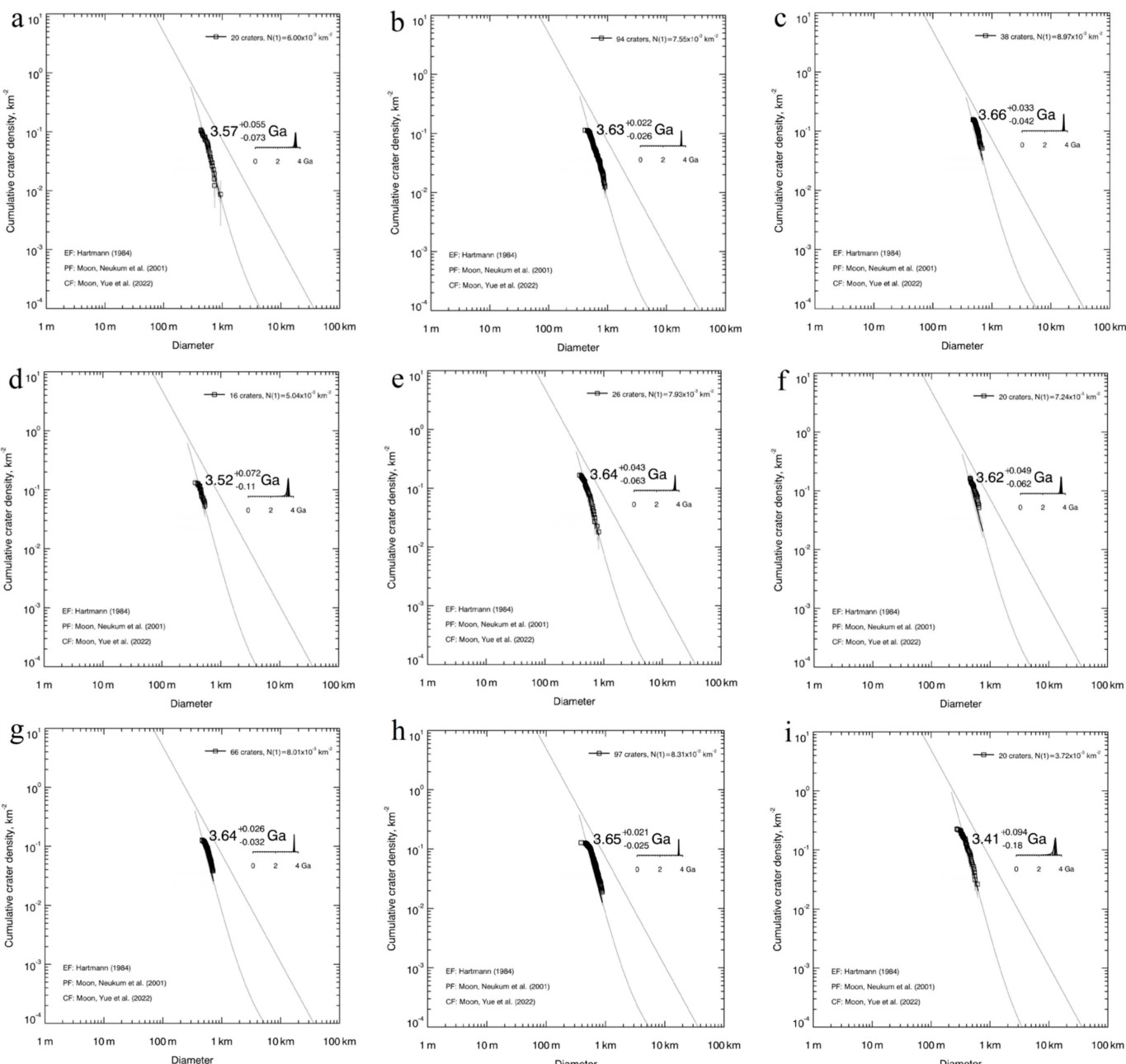

**Figure 6.** Model ages of units [23–25] (**a**) Model ages of unit A. (**b**) Model ages of unit B. (**c**) Model ages of unit C. (**d**) Model ages of unit E. (**e**) Model ages of unit F. (**f**) Model ages of unit G. (**g**) Model ages of unit H. (**h**) Model ages of unit I. (**i**) Model ages of unit J.

**Table 2.** Model ages of Tsiolkovsky mare basalts.

| Unit Name | Dating Area (km$^2$) | Number of Craters | Model Ages (Ga) | Deviation (Ga) |
|---|---|---|---|---|
| A | 283.08 | 20 | 3.57 | +0.055/−0.073 |
| B | 1259.40 | 94 | 3.63 | +0.022/−0.026 |
| C | 348.97 | 38 | 3.66 | +0.033/−0.042 |
| E | 254.94 | 16 | 3.52 | +0.072/−0.11 |
| F | 222.21 | 26 | 3.64 | +0.043/−0.063 |
| G | 187.73 | 20 | 3.62 | +0.049/−0.062 |
| H | 775.10 | 66 | 3.64 | +0.026/−0.032 |
| I | 1020.44 | 97 | 3.65 | +0.021/−0.025 |
| J | 195.60 | 20 | 3.41 | +0.094/−0.18 |

### 4.5. Structural Characteristics

In this paper, the structural features of the Tsiolkovsky crater are interpreted and analyzed with reference to the 1:2,500,000–scale global tectonic map of the Moon by Lu Tianqi et al. [14]. Two types of linear structures were developed in this region: impact fractures and wrinkle ridges (see Figure 7). In the study area, impact fractures with lengths of 16–104 km developed simultaneously with the crater in the late Imbrian epoch. They mainly occurred on the wall of the Tsiolkovsky crater, basically parallel to the crater wall, and exhibited a terraced-wall pattern. The colors on both sides of the impact fault in the false-color image were quite different, and there was an obviously steep landslide. It can be concluded that these impact fractures were formed by both impact and collapse.

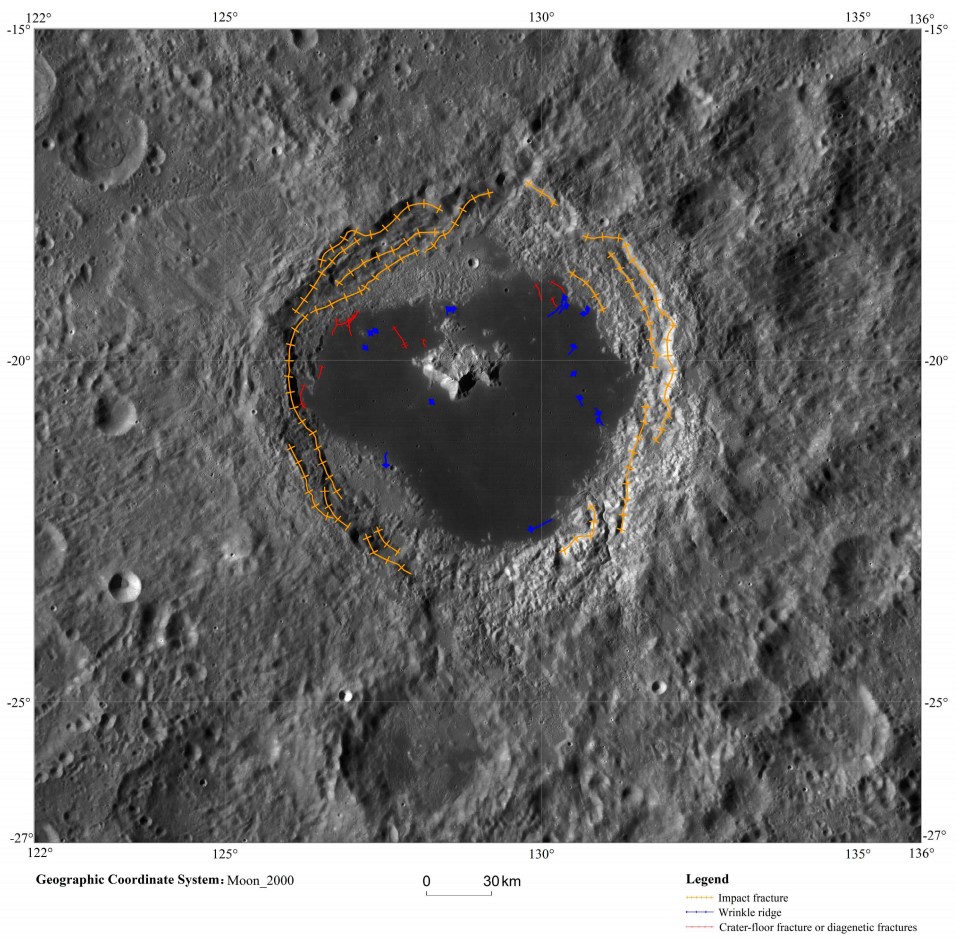

**Figure 7.** Tectonic map of the Tsiolkovsky region.

Wrinkle ridges with lengths ranging from 1 km to more than 10 km and a maximum length of 13.73 km were mapped. They developed on the basalt units at the floor of the Tsiolkovsky crater and formed later than the basalt, mostly due to the horizontal contraction of magma during cooling [36,37]. In addition to the above two types of structural features, dozens of fractures are observed in the floor of the crater. Most of the fractures are located at the edge of the crater floor with only two located near the central peak, radially pointing to the central peak. These fractures are vague and short in length (several hundred meters). These rocks may be modified by mare basalt and are speculated to constitute crater floor fractures formed by volcanic intrusion or diagenetic fractures formed by magma condensation and contraction. These features are further discussed in Section 5.2.

## 5. Discussion

### 5.1. Stages and Characteristics of Regional Volcanic Activity

Although flat in topography and with a uniformly dark albedo, the compositional variations in the Tsiolkovsky crater floor suggest that the region may have undergone a more complex volcanic history. According to the statistical results of chemical composition and age, the nine basalt units at the crater floor could be divided into two categories (see Figure 8). The surface age of the Tsiolkovsky crater floor is between 3.66 and 3.41 Ga, i.e., late Imbrian. The compositional statistics of each episode of volcanic activity were obtained and compared with those of the Apollo, Luna, and Chang'e-5 samples (see Table 3 and Figure 9).

1.  The first episode of volcanic activity occurred at 3.5–3.7 Ga, and multiple lava flows erupted in the Tsiolkovsky crater floor with a time span from 3.66 Ga to 3.52 Ga and a duration of ~331 Ma. The total exposed area of mare basalts that formed in this episode is 11,854 km$^2$. Compared with those of the Apollo, Luna, and Chang'e-5 samples, the magma compositions that erupted in this region are similar to those of the basalt samples from the Apollo 17 and Apollo 14 landing sites. The compositions of these eruptions are highly evolved, i.e., with higher FeO and HCP contents and lower Al$_2$O$_3$ content and Mg# value of mafic minerals.

2.  The second episode of volcanic activity erupted at 3.41 Ga, and a small eruption occurred in the northernmost part of the Tsiolkovsky crater floor with an area of 234 km$^2$. The major element composition is similar to that of the Apollo 16 basalt samples; was less evolved; and is characterized by the lower contents of FeO and TiO$_2$, the higher contents of Mg# value of mafic minerals and Al$_2$O$_3$, and the spectral signature of low–calcium pyroxene.

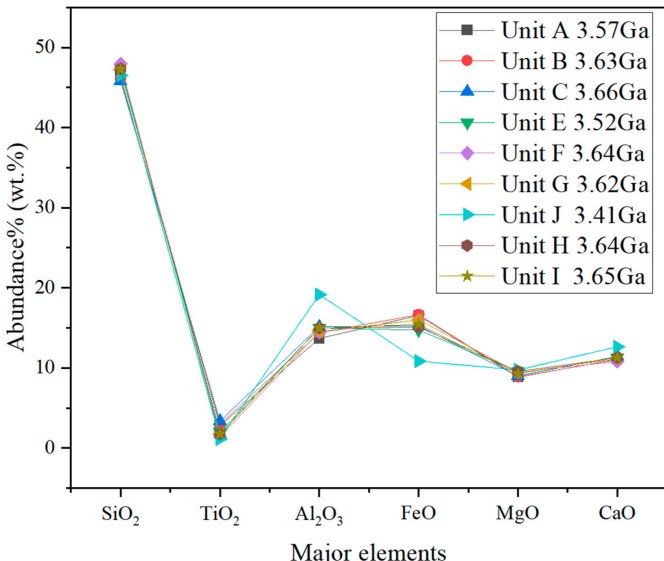

**Figure 8.** Major element abundance of Tsiolkovsky basalt units.

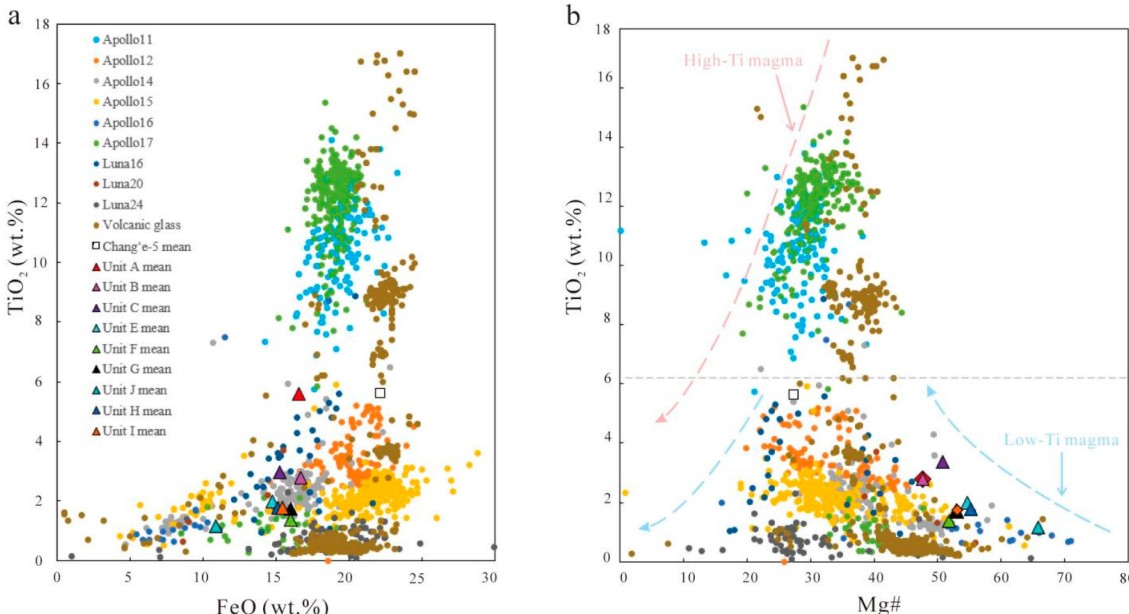

**Figure 9.** Subfigure (**a**,**b**) are comparison of geochemical compositions between Tsiolkovsky and returned samples of mare basalts. Data of Apollo and Luna basalts are modified from Neal [38] and Zhao et al. [22]. Data point of Chang'e-5 basalts is from Yang et al. [1]. In panel (**b**), some points of Tsiolkovsky units are overlapped, and the symbols of units A and I are changed to squares for distinguishment.

**Table 3.** Model ages and average compositions of the basalt units. The unit of Th abundance is ppm, and the units of the other elements are wt.%. Mg# values are unitless.

| Model_Age | FeO | TiO₂ | Mg# | Th | Al₂O₃ | CaO | SiO₂ |
|---|---|---|---|---|---|---|---|
| ~3.4 Ga | 10.84 | 1.15 | 65.89 | 1.35 | 19.14 | 12.64 | 46.5 |
| 3.5–3.7 Ga | 15.71 | 2.22 | 51.73 | 1.28 | 14.63 | 11.17 | 47.03 |

The duration of each episode was not long, and the compositions of each unit that formed in the same episode are similar. Based on the analysis of the age and geological characteristics, the volcanic activity in this region should include small and multistream eruption events.

### 5.2. Regional Evolution Process

Based on the current knowledge, the surface of the Moon was entirely molten at the beginning of its formation approximately 4.5 Ga ago, forming a "magma ocean" with a depth of several hundred kilometers. With decreasing temperature and fractional crystallization of the magma ocean, the Moon underwent complete crust–mantle differentiation. The widely accepted model of lunar "magma ocean" differentiation suggests that the early crystallized dense olivine and pyroxene crystals sank to form the cumulates of mantle peridotite. Until the magma ocean was approximately 60~80% solid, the low–density plagioclase crystallized and floated upwards, forming the primary lunar crust of anorthosite on the surface of the lunar magma ocean [39,40]. The Fermi basin formed at 4.05 ± 0.1 Ga [34], when the lunar crust was thinned and modified by the impact, and a large amount of melt may have been generated. During the late Imbrian (<3.8 Ga), the Tsiolkovsky impact again thinned and altered the lunar crust in this region.

Between 3.66 and 3.41 Ga, volcanic events dominated the geological evolution of the region with lava flows erupting at the Tsiolkovsky crater floor to form mare plains. The compositional and chronological characteristics of volcanic activity during each episode are described in detail in Section 5.1. The FeO, TiO₂, and Al₂O₃ contents of the Tsiolkovsky

basalts were compared with those of returned samples (see Figure 10). Based on Neal and Taylor's classification [41], the nine units in the study area are all low-titanium basalts. The gradual accumulation of heat released by the decay of radioactive elements led to partial melting of the cumulus material, after which the magma was separated and fractionally crystallized during the upwelling process before finally flowing through the channel to the lunar surface to form the mare basalt in this area.

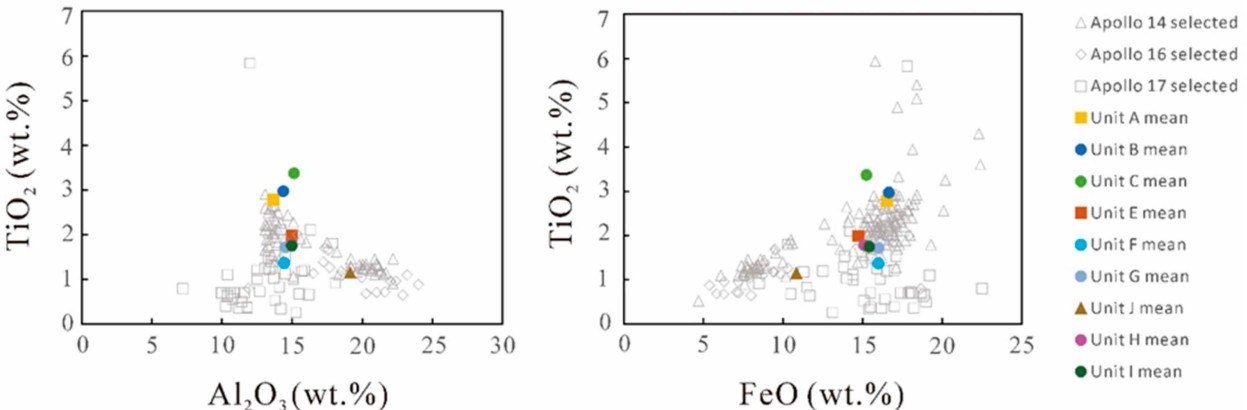

**Figure 10.** Comparison of Tsiolkovsky mare basalts and returned samples with similar components.

Magma eruption to the lunar surface required channels. In Section 4.5, we identified blurry fractures that developed at the crater floor and speculated that these fractures may be caused by magma intrusion [42,43] or by condensation and contraction of lava flows, similar to the diagenetic fractures that occur on Earth.

If the fractures are crater floor fractures, then the mare basalt may have formed by magma flowing up along the fractures in the lunar crust to the lunar surface under the action of the diapir, and the crater floor experienced a large area of mare basalt flooding and even filled up many previously formed crater floor fractures. This viewpoint is consistent with those of Baldwin et al. [44] and Walker and El-Baz et al. [6]. If the fracture was a diagenetic fracture, then there may have been two eruption channels in the mare basalt: one was the central peak, and the other was the impact fracture. However, there was an odd occurrence: the Fermi basin, which is larger than the Tsiolkovsky crater, also produced a central peak, but there are no mare basalts on its floor. We propose three hypotheses for this discrepancy: first, the Fermi basin once erupted basalt, but it was covered by ejecta materials from the Tsiolkovsky crater, becoming a cryptomare; second, there was no magma reservoir below the impact location of the Fermi Basin; third, there may have been magma reservoirs below the impact site of the Fermi basin, but the lunar crust was less brittle in the Aitkenian and did not develop fractures, and magma could not rise to the lunar surface. The Moon was in a state of continuous cooling, and in the late Imbrian, when the Tsiolkovsky impact event occurred, the lunar crust was relatively brittle, thus forming impact fractures in the crater wall and providing channels for the eruption of basalt. This hypothesis is similar to that of Wilbur et al. [45], who theorized that the mare basalts at the Tsiolkovsky impact crater floor were the result of several small volcanic events rather than a single volcanic flooding event in which magma from several sources erupted into the lunar surface along channels constructed through the crater floor and the crater wall. Moreover, they further proposed that faults and central peak uplift provided channels for the magma source region and that faults were more important.

## 6. Conclusions

The topography, compositions, and ages of the basalt units in the Tsiolkovsky crater were analyzed, and the structural features were interpreted and studied in detail. According to the compositional data, nine basalt units were identified, and their model ages were

obtained. The eruption ages span from 3.66 Ga to 3.41 Ga. We classified them into two episodes of volcanic activity with maxima occurring at 3.5–3.7 Ga and ~3.4 Ga. These basalts are low–titanium basalts, and their Mg# values are higher than those of the Chang'e-5 samples, indicating a relatively low degree of volcanic evolution. These basaltic magmas were partially melted from the lunar mantle along crevasses in the lunar crust or from the central peak and impact fracture and then erupted to the lunar surface with fractional crystallization.

This paper provides a detailed geological analysis on the volcanic evolution of the Tsiolkovsky crater. This study could provide a case study for other impact craters where basalt is exposed on the farside of the Moon. Future research will be performed on these basalts as more information regarding the basalts on the farside of the moon becomes available (e.g., the Chang'e-6 samples).

**Author Contributions:** Conceptualization, Y.W. and X.D.; methodology, Y.W.; software, Y.W. and J.C.; validation, Y.W. and X.D.; formal analysis, Y.W.; investigation, Y.W.; resources, Y.W.; data curation, Y.W.; writing—original draft preparation, Y.W.; writing—review and editing, Y.W., X.D., J.C., K.H., C.S., M.J. and J.D.; visualization, Y.W., L.L. and X.L.; supervision, X.D.; project administration, X.D.; funding acquisition, X.D. All authors have read and agreed to the published version of the manuscript.

**Funding:** This research was funded by the Geological Survey Projects of China Geological Survey (Grant No. DD20221645) and the National Science and Technology Infrastructure Work Projects (Grant No. 2015FY210500).

**Data Availability Statement:** The SLDEM2015 is available from LOLA Planetary Data System (PDS) Data Note (http://imbrium.mit.edu/EXTRAS/, accessed on 30 August 2020). The LRO WAC images are available from the LRO data collections. https://astrogeology.usgs.gov/search/map/Moon/LRO/LROC_WAC/Lunar_LRO_LROC-WAC_Mosaic_global_100m_June2013, accessed on 3 February 2020). The thorium abundance data are available from the PDS Geoscience Node (https://pds-geosciences.wustl.edu/missions/lunarp/reduced_special.html, acceseed on 30 August 2020). The MgO abundance data, the $Al_2O_3$ abundance data and the Cao abundance data are available from the repository Zenodo via DOIs: https://doi.org/10.5281/zenodo.7263425, https://doi.org/10.5281/zenodo.7262473 and https://doi.org/10.5281/zenodo.7263324.

**Acknowledgments:** We are grateful for the LRO series data provided by the Planetary Data System. And we also thank those who worked on the Chang'e samples to obtain composition. We also appreciate the constructive feedback from anonymous editors.

**Conflicts of Interest:** The authors declare no conflicts of interest.

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
