# Peer review of "Interpretation of Geological Features and Volcanic Activity in the Tsiolkovsky Region of the Moon"

_remotesensing, doi:10.3390/rs16061000_

Round 1
Reviewer 1 Report
Comments and Suggestions for Authors
Dear authors,
I read the article "Interpretation of geological features and history of magmatic activity in the Tsiolkovsky region of the Moon" and I understand the scientific relevance of the study. In my opinion, the article is well structured, with a well-described methodology and results that support the discussions made. I have made some improvements, mainly in the introductory part of the article. Although I saw scientific relevance in the study, I have made some suggestions throughout the text which I believe can better present the relevance of the topic and the challenges associated with remote sensing of celestial bodies that your work helps to address. Some of the figures in the article need improvement. I have highlighted these figures in my suggestions. In some cases, it is impossible to see the information that is being presented in the image. I believe that if the improvements are made, the article will be up to the journal's standards and will bring a lot of discussion to the scientific community.

Comments on the Quality of English LanguageI didn't have any major problems with the quality of the English. Although I'm not a native speaker, it was very easy to read.
Author Response
We sincerely appreciate the time and thoughtful comments devoted by the reviewers. We have carefully revised our manuscript according to the reviewers’ comments and suggestions. The detailed responses to the comments are given below. The corresponding revisions have also been highlighted in yellow color in the manuscript.

Reviewer 2 Report
Comments and Suggestions for Authors
The paper deals with the 'magmatic' activity in the Tsiolkovsky region of the Moon. As such the paper is well written and presents an important scientific account of the mare basalts exposed in the crater. I wish to make the following suggestions/comments:
1. I have only one suggestion to make- since the basalts are exposed on the surface of the crater flow the term should be 'volcanic' rather than 'magmatic' as the later gives a subsurface connotation. So, the word 'magmatic' should be replaced by 'volcanic' in the title and at relevant places in the text.
2. In Table 3, the model ages for basalts (~3.5 Ga and `3,6 Ga) may be different, but compositional they have almost similar geochemistry? what are the implication for this in terms of partial melting? does it mean that the sample mantle (cumulate peridotite as mentioned in text) remelted and what caused the two episodes of melting? (direct impact elsewhere vs. decompression melting due to fracturing?)
3. In the conceptual model (Fig. 11) a lot needs to be explained. First the crater is shown exclusively in the lunar crust? was the crust so well developed at ~4.5 Ga?? secondly, what is the magmatic plumbing system (with vents, sills and dykes doing in the model?) The lower half of the model looks like a hotspot (plume?) completed with a well-developed magmatic plumbing system as seen in komatitite volcanism on early Earth. The upper half of the model looks like an impact crater coincidentally located on this hotspot? I don't think the authors want to depict that sort of a speculative model. Surely a better model drawn based on the descriptions in the text.
All the best.
Author Response

(The authors gave the same response as above.)

Reviewer 3 Report
Comments and Suggestions for Authors
Comments on ‘Interpretation of geological features and history of magmatic activity in the Tsiolkovsky region of the Moon’ by Ying Wang et al.
I have read the manuscript with much interest and find that this would be important attempt and contribution. Below I list my three concerns and hope that the authors may consider them properly.
1. The motivation of the ms is not very strong, it is not very clear in the introduction why studying the Tsiolkovsky region is of great importance, surely not that this region has not been comprehensively studied by geologists before.
2. Please indicate more clearly the the chemical positions are derived from the methods like the WAC, and how topographic and geological data are reconciled, they sometimes look confusing.
3. The authors may discuss a broader selenology significance using this case study.
Regards
Feb 18, 2024
Comments on the Quality of English LanguagePlease have a check of the typos.
Author Response
We sincerely appreciate the time and thoughtful comments devoted by the reviewers and the associate editor. We have carefully revised our manuscript according to the reviewers’ comments and suggestions. The detailed responses to the comments are given below. The corresponding revisions have also been highlighted in yellow color in the manuscript.

Round 2
Reviewer 2 Report
Comments and Suggestions for Authors
This is a good contribution to studying the far side of the moon.
Comments on the Quality of English LanguageThe English is generally good, but some improvement can be made to tighten the text.
Reviewer 3 Report
Comments and Suggestions for Authors
I have no further concern, the ms is now acceptable.
Zhiyuan He, PhD
Ghent University
28 Feb, 2024
Author Response
Thank you for your comments and guidance on my article. Please feel free to contact me if you have any questions. Wish you a happy life.